# Assessment of Asteroid Classification Using Deep Convolutional Neural Networks

Victor Bacu *, Constantin Nandra, Adrian Sabou, Teodor Stefanut and Dorian Gorgan

Computer Science Department, Technical University of Cluj-Napoca, 400114 Cluj-Napoca, Romania;
constantin.nandra@cs.utcluj.ro (C.N.); adrian.sabou@cs.utcluj.ro (A.S.); teodor.stefanut@cs.utcluj.ro (T.S.);
dorian.gorgan@cs.utcluj.ro (D.G.)
* Correspondence: victor.bacu@cs.utcluj.ro

**Abstract:** Near-Earth Asteroids represent potential threats to human life because their trajectories may bring them in the proximity of the Earth. Monitoring these objects could help predict future impact events, but such efforts are hindered by the large numbers of objects that pass in the Earth's vicinity. Additionally, there is also the problem of distinguishing asteroids from other objects in the night sky, which implies sifting through large sets of telescope image data. Within this context, we believe that employing machine learning techniques could greatly improve the detection process by sorting out the most likely asteroid candidates to be reviewed by human experts. At the moment, the use of machine learning techniques is still limited in the field of astronomy and the main goal of the present paper is to study the effectiveness of deep convolutional neural networks for the classification of astronomical objects, asteroids in this particular case, by comparing some of the well-known deep convolutional neural networks, including InceptionV3, Xception, InceptionResNetV2 and ResNet152V2. We applied transfer learning and fine-tuning on these pre-existing deep convolutional networks, and from the results that we obtained, the potential of using deep convolutional neural networks in the process of asteroid classification can be seen. The InceptionV3 model has the best results in the asteroid class, meaning that by using it, we lose the least number of valid asteroids.

**Keywords:** image classification; astronomy; asteroids; convolutional neural network; deep learning

## 1. Introduction

Near-Earth Objects (NEOs) and especially Near-Earth Asteroids (NEAs) can pose a clear threat to human life and property because of their proximity to the Earth. By predicting potential future impacts with our planet [1], decision-makers could be informed of the danger, which would offer them time to start working on damage mitigation strategies. Improving the prediction chances requires the constant surveying of the nearby space around the Earth in order to continuously monitor for NEOs and NEAs and also to discover previously unknown objects. This requires the analysis of large datasets of telescope images, which can quickly add up during observation nights. After identifying potential asteroid candidates from the image data, the results are sent to human observers to review and confirm them. In this context, the timely and effective processing of the data is extremely important, as it provides the human observers with a steady stream of objects for review. However, it is also essential that the number of candidate objects does not overwhelm the human resources available. To this end, we decide to employ machine learning techniques in an attempt to improve the object detection rate. Coupling artificial intelligence with the use of high-performance distributed processing infrastructures such as cloud-based solutions—which have seen widespread adoption as of late following the constant increase in computational and storage power—we think that we could maximize the benefits of having access to massive volumes of data in the field of astronomy.

This study is conducted within the scope of the CERES project [2] aiming at designing and implementing a software solution that is capable of classifying objects detected in

astronomical images. The focus is on detecting/identifying asteroids. To accomplish this objective, we rely on machine learning techniques to build up an asteroid classification model. It is important to minimize the false-negative results, meaning asteroids that are incorrectly classified. This model is used in conjunction with another software tool that detects valid asteroid trajectories. Therefore, even though the classification model produces some false positives, our trajectory validation tool eliminates these detections as they should not be part of valid trajectories. On the other hand, if the classification model generates false negatives, this means that we miss these asteroids.

To accomplish the objective of building up an asteroid classification model, we need to create a dataset of asteroid images and train a model to be used for the classification of new objects. Figure 1 shows the conceptual description of the CERES project. The dataset consists of series of subsequent images of the same asteroids. The training of the asteroid classification model relies on the dataset of asteroid images and other modules rely on the inference part, meaning the employment of the trained model to make predictions. Within this study, asteroid classification refers to the recognition of asteroids amidst various other objects found in astronomical images, such as stars, galaxies, and noise.

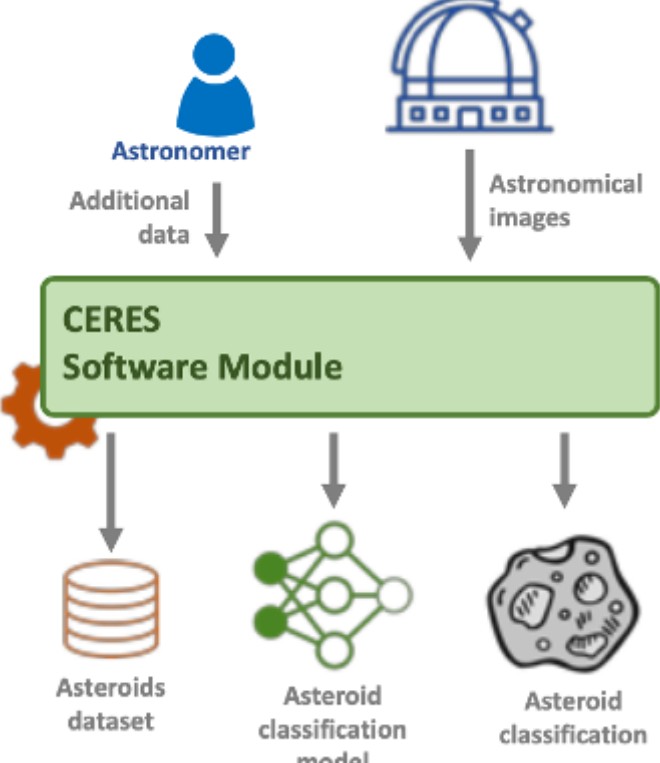

**Figure 1.** Conceptual description of the CERES project.

The main goal of the present paper is to review the effectiveness of deep CNNs at the task of classifying astronomical objects, specifically asteroids. We present our findings by comparing the results obtained from some of the most well-known deep convolutional neural networks (CNNs), including InceptionV3, Xception, InceptionResNetV2 and ResNet152V2. These state-of-the-art classification CNNs are used to explore the suitable direction to this particular classification problem, either by full training or by fine-tuning.

This paper is structured as follows: Section 2 presents some related works on the presented topic, while Section 3 presents some background notions on convolutional neural networks. The next sections describe the methodology used to assess various CNNs in the process of asteroid classification while describing and discussing the experiments and the obtained results. The last section concludes the ideas presented within this paper.

## 2. Related Works

Machine learning is a technique that simulates the way humans learn—through examples—and is used to develop software applications capable of deducing information from input datasets by means of generalization. This ability of generalization is developed and continuously improved by performing repeated analysis of some examples, also called training data. After this analysis (or training) is conducted, the machine-learning-based applications can apply the learned generalization rules to infer results from new datasets which were not used in the training phase. One of the most well-known and used machine learning techniques employs an abstract model that emulates the functioning of our brain based on an attempt to simulate the interactions that take place between neurons. Although the representation model of a neuron is not new, as it was proposed more than half a century ago [3], the applicability of neural networks in solving real problems was limited due to the limited processing power of computers and, to a lesser extent, the availability of datasets needed to train such models.

Typically, problems that can be solved by applying machine learning techniques are grouped into two major categories: supervised and unsupervised learning problems. In the case of the former, the data are already annotated with various labels. The unsupervised learning process is characterized by the lack of labels or other attributes that could help in the classification task. In this case, the learning algorithms must identify possible structures, groups or trends within the provided datasets. A supervised learning process should lead to the identification of some dependency relationships between the input dataset and a specified result. This relationship is obtained after feeding the system with a large number of examples and it can afterwards be used to predict the outcome for a new dataset.

CNNs are common methods in the image processing literature and could be applied for various fields. No-reference image quality assessment by exploiting convolutional neural networks and other deep learning techniques is presented in [4]. Transfer learning could be used with success in medical imaging [5,6].

In the field of astronomy, researchers are capturing various sets of images using different telescopes. Most of the time, the data are obtained after repeated sessions of observations of areas of the sky (fields), which can lead to the accumulation of large volumes of data that need to be analyzed in order to extract useful information [7]. To make effective use of such data, classification is one of the most widely seen applications of machine learning techniques, for example, in [8]. In this context, most classification use cases rely on neural networks with multiple hidden layers of neurons, also called deep networks. The use of such deep neural networks is made possible and accessible by the evolution of hardware [9] and of software capable of running such models effectively [10].

Asteroids are being discovered by sky surveys. Ref. [11] presents a comprehensive debiasing Bayesian framework applicable to brightness-limited surveys of solar system objects. Investigation of the capabilities of neural networks for the automated classification of asteroids based on their taxonomic spectral classes is presented in [12]. Ref. [13] analyzes meteorites through mid-infrared spectroscopy and considering possibilities for thermal infrared observations in the context of the binary asteroid Didymos.

A good example that illustrates the kinds of problems that have to be solved due to the increasing amounts of data generated by telescopes (with high resolution and covering a large field of view) is the Galaxy Zoo project [14]. During this project's life cycle, the task of classifying 900,000 galaxy images was distributed to a group of volunteers. This is exactly the type of problem that can be simplified by migrating the repetitive task of classifying images from astronomers to computers running neural network classifiers. Such a solution is proposed by the authors of [15], demonstrating the feasibility of classifications performed using a neural network trained on a dataset that was previously labeled by volunteers. Experimenting with various combinations of parameters, the results obtained using the proposed neural network reached a level of accuracy of 90% when compared to the classifications performed by humans volunteers.

The two major directions of research related to the identification of planets outside of the Solar System (exoplanets) and asteroids in the Solar System or near Earth are relevant examples of classification problems. Furthermore, within this domain, we can mention also the automated detection of transient objects such as supernovae, novae, variable stars, and even artificial satellites, all of which require separation from other entities. In [16], the authors describe a method for detecting exoplanets based on multi-level neural networks. The model was trained using data from the Kepler space telescope to identify Earth-sized exoplanets by the transit method. This implies measuring the variation of a star's light, variation that is a direct result of an exoplanet's transit between the star and the observer. The paper highlights the challenges of identifying such small planets mainly due to the noise which, at that scale, is difficult to differentiate from the valid signals. This problem of identifying exoplanets when having low values for the signal-to-noise ratio is also addressed in [17–19], where the authors present the results of using convolutional neural networks to differentiate between valid signals and noise caused by other astronomical phenomena or instrument limitations.

Another interesting classification example in the field of astronomy is for the purpose of detecting potentially habitable exoplanets [20]. With the number of discovered exoplanets expected to reach the order of millions in a few years, the author presents some machine learning models that could be useful in identifying potential candidates similar to Earth. These models take into account the characteristics of the candidate planet and the star around which it orbits, one of the models even providing an index of similarity to our planet.

The authors of paper [21] present a study on using different neural network architectures for star classification. The study includes models such as recurrent, dilated and temporal convolutional neural networks, LSTM (long short-term memory) cells and autoencoders. The results suggest the superiority of recurrent networks in terms of accuracy, while convolutional networks have the advantage when it comes to minimizing training time and memory usage.

The detection and classification of Near-Earth Asteroids (NEA) is another area of interest in astronomy, especially given the danger that these objects pose due to the damage they could cause in the event of an impact [22]. This is one of the main reasons for NASA's attempts to identify, track and catalog NEAs, in order to make predictions on potential impact hazards [23].

In the literature, one can distinguish two main directions of research related to the detection and classification of NEAs: analysis and classification based on the trajectory attributes of asteroids and their detection from images obtained by recurrent observations. The authors of [24] describe a solution to use machine learning techniques to predict the orbital parameters of asteroids. The proposed method is based on an SVM (Support Vector Machine) algorithm to identify potentially dangerous subgroups of asteroids that are found in major NEAs groups. The authors of [25] present the results of using a neural network to identify asteroids that can present an impact risk. As a training dataset, the solution uses simulated trajectories that are obtained by the inverse integration in time of some objects launched from the Earth's surface. The results presented indicate an identification rate of up to 90% of all the objects that present an impact risk.

Identifying NEAs from sets of images obtained from telescopes is the main method of discovering new asteroids. After the initial discovery, the asteroids are confirmed and tracked periodically, while their orbits are being modelled. All these data are stored in an international database, the Minor Planet Center [26]. Starting from these data, different studies can be performed on the existing NEA populations and the risk of impact that they present. NEOWISE [27], Catalina Sky Survey [28], ATLAS [29] and LINEAR [30] are projects dedicated to identifying new asteroids. The data produced by these projects are far beyond the processing capabilities of human observers. In [31], the authors present a machine learning solution utilized to classify the detected objects from NEOWISE images. The experiments used the Python library sklearn to define models and datasets validated by astronomers for training. In that dataset, the classes distinguish between asteroids and

other identifiable objects, such as galaxies, cosmic rays or instrument artifacts. Similar solutions based on the use of convolutional neural networks to identify asteroids are presented in [32–34].

The authors of [35] present their achievements in creating a real-time NEA discovery pipeline that uses a machine learning classifier to filter a large number of false-positive streak detections. It offers to the human scanner an effective and remote solution able to identify real asteroid streaks during the night. The machine learning classifier is based on a supervised ensemble-method approach for classification, namely Random Forest. Another approach based on machine learning to detect moving objects is presented in [36]. Here, the features used for building the model are photometry measurements, position measurements and shape moments. For the purpose of asteroid identification, the motion observed between consecutive images becomes relevant when contrasted with the stationary appearance of stars.

The Near-Earth Asteroid Tracking (NEAT) program was a pioneer in implementing a fully automated system for detecting asteroids. This included controlling the telescope, acquiring wide-field images and detecting NEOs using onsite computing power provided by a Sun Sparc 20 and a Sun Enterprise 450 computer [37]. One approach that maintains high effectiveness while producing low false detection rates involves using two independent detection algorithms and combining the results, reducing the operator time associated with searching huge datasets. This solution is described by Petit et al. [38], their efforts having resulted in the creation of a highly automated software package for the detection of moving objects.

## 3. Background on CNNS

### 3.1. Deep Convolutional Neural Networks

The architectural design of Convolutional Neural networks (CNN) is based on multiple interconnected layers, where different features (low, intermediate or high) are sequentially extracted. The main building blocks of every CNN are the convolution and pooling layers, followed by some fully connected layers, the final layer being typically based on the softmax function [39]. Features are learned by the convolution and pooling layers, whereas fully connected layers and softmax functions are responsible for the final classification. Based on some filtering functions, the convolutional layers extract various spatial features from the images. In the case of asteroid classification, these features define the typical geometrical representation of an asteroid and the patterns that are specific to asteroids. This feature extraction method is performed hierarchically, meaning that the first convolutional layers learn low-level features, such as edges or corners, while the last layers extract high-level features.

At its most basic level, a convolution is a simple mathematical operation with the image matrix and a particular filter as inputs. The convoluted features are obtained by sliding a fixed-size window over the input data matrix and applying some filter. The stride specifies the shift in the pixels of the sliding window. One of the most widely used activation functions in this type of approach is the Rectified Linear Unit (ReLU) [40]. Even though it is a simple function, linear for positive values and zero for negative values, it yields good results in various applications of CNNs. As it is cheap to compute, it does not have a great impact on the training time, which could increase with the size of the dataset. Pooling layers are often defined after each convolutional layer and they reduce the dimensionality by sub-sampling or down-sampling. These layers capture the presence of distinct features, patterns, or textures in the image. This helps in maintaining the relevant features while discarding less significant details. Such a layer is based on simple mathematical operations, like maximum or average calculation. The operations are performed in a manner similar to that of the convolutional layers, based on a sliding window. The difference comes from the stride value, which is higher than one, so that the result is a down-sample of the feature map. By reducing the dimensionality of the feature maps, the computational requirements are also reduced.

Between the classification layer and the convolutional/pooling layers, some CNNs could have fully connected layers. The presence of such layers could lead to overfitting. Overfitting is a frequent issue in machine learning, where a model becomes hyper-focused on learning from the training data to such an extent that it not only captures the fundamental patterns, but also absorbs the noise and random fluctuations present within the data. This can be mitigated by using dropout layers. By selectively deactivating neurons along with their corresponding connections in a random manner, we avoid the formation of interdependence among neurons throughout the training process. The classification layer, the last one in the architecture, typically the softmax, converts the output of the CNN into a set of probabilities, which add up to one. This output represents the probability of the input being a member of a particular class. In a CNN architecture, connections between different layers play a crucial role in enabling the network to learn hierarchical features and make accurate predictions. These connections facilitate the flow of information and transformations from the input layer through intermediate layers to the final output layer.

### 3.2. Transfer Learning

By using a well-known deep CNN, we rely on a pre-trained model that contains all the weights representing the features identified on the dataset used to train this model. In this study, we seek to determine whether these learned features are transferable to a different dataset, namely to astronomical images. The other objective of this study is to identify the most suitable deep CNN to be used in the process of asteroid classification.

Transfer learning is a method in which we are reusing features learned on a given problem to solve a new one. It is usually performed when our dataset has too little data and we transfer the knowledge gained by training on a large dataset. A typical transfer learning workflow consists of the following steps:

- Start from the layers of the previously trained model;
- Freeze their weights in order to maintain what the model learned during training on a large dataset;
- Add new layers on top of the previously trained model but make them trainable;
- Train the new model, basically the newly added layers.

Our dataset of asteroid/non-asteroid detections is relatively small (number of images) when compared to other datasets (such as ImageNet). With this in mind, fine-tuning using such pre-trained CNN seems to be the right approach. Fine-tunning means making adjustments within the top layers of the CNN while freezing the bottom layers that are responsible with low-level features (such as edges or corners). During the training phase, the weights of the layers that were previously frozen are not updated. By using this approach, we retain some of the weights of the pre-trained model on a particular dataset and at the same time we are making some adjustments in order to accommodate the new dataset. The top layers are responsible with identifying generic features, while the bottom layers are responsible with identifying specific features of the studied dataset. If we were to conduct fine-tunning on all of the layers, it could lead to an overfitting situation.

Starting training from scratch (full training) is a better option when the new dataset is sufficiently large to enable the model to reach convergence. Obviously, the computational cost of this approach is much higher than that of the the fine-tunning approach. This involves taking a pre-trained model that has already learned useful features from one dataset and further refining it for a new dataset. Often, in the case of full training, we could encounter an overfitting situation, and in order to mitigate this, we would add some dropout layers or try to augment the data. Having a small dataset would not help the model to properly generalize. We could rely on data augmentation to increase the dataset size. The data augmentation would consist of some transformations, including translation, rotation and reflection. These would help to increase the size of the dataset. For our particular dataset (binary set), data augmentation is not a good idea because it would increase the number of images that are labeled as non-asteroids but have the same particularities as asteroids.

## 4. Methodology

In this study, we start by using four state-of-the-art CNNs to classify asteroids in astronomical images. Figure 2 highlights the main building blocks of the classification model. We add some more layers to the base layers of the used CNNs. In this section, we present the four CNNs on which we test the asteroid classification use-case. These are InceptionResNetV2, InceptionV3, Xception and ResNet152V2, and they are chosen based on their already confirmed performance in classification use-cases. We start by briefly describing them and highlighting the changes that we introduce on these models.

In order to avoid overfitting, we use the early stopping method. Throughout training, the model is evaluated on the validation dataset at the end of each epoch. We stop the training process if the performance of the model on the validation dataset starts to degrade. This means that the validation dataset loss starts to increase or the validation dataset accuracy starts to decrease. Such a model should have a good generalization performance.

### 4.1. InceptionResNetV2

This CNN network integrates two well-established deep CNNs, ResNet and Inception, both displaying good performance and maintaining a low computational cost. Instead of the filter concatenation stage, this model includes residual connections into the Inception architecture. Residual connections involve creating a direct shortcut between earlier and later layers. This shortcut allows the bypass by the original input of certain intermediate transformations. This allows for an increase in the number of inception blocks, and obviously in the network depth, but without compromising the computational cost. The architecture consists of 164 layers [41]. The architecture is built on a foundation of Residual connections and Inception structures. Multiple convolutional filters of various sizes are mixed with residual connections in the Inception–Resnet block. Images with three channels and a resolution of $299 \times 299$ serve as the model's input data. The architecture starts with a few layers of convolutions that increase the number of channels while decreasing the spatial dimension. The training duration is shortened by using residual connections in the subsequent blocks of Inception–Resnet and Reduction, which learn the features. Average pooling, drop-out, and soft-max are the last layers.

### 4.2. InceptionV3

GoogLeNet is one of the first nonsequential CNNs. By increasing the depth and width of a deep CNN model, we can improve the performance, but at the cost of also increasing the number of parameters. The presence of many parameters can impact the computational cost. To tackle this problem, the Inception module [42] was introduced in GoogLeNet. This module is based on $1 \times 1$, $3 \times 3$ and $5 \times 5$ convolutions followed by a filter concatenation. This solution can reduce the number of training parameters and that, in turn, can further reduce the computational complexity.

### 4.3. Xception

The Xception architecture is built up as a linear stack of depth-wise separable convolution layers with residual connections. The 36 convolutional layers act as a feature extraction base for this network. There are 14 modules made up of the 36 convolutional layers. With the exception of the first and last modules, all modules have explicit linear residual connections. Entry flow, middle flow, and exit flow are the three sections of the Xception architecture. The entry flow is the initial step the data takes before moving through the middle flow. It consists of several Convolution and SeparableConvolution layers. The middle flow which is repeated eight times is composed of SeparableConvolution layers. Lastly, the exit flow concludes this architecture [43].

### 4.4. ResNet152V2

Within the ResNet [44] family, we find multiple deep neural networks that share a similar architecture, the difference being in the depth of the model. ResNet is based on

residual learning units, with the objective of reducing the degradation of deep neural networks. CNNs of this category are preferable because they typically produce good classification accuracy without increasing the model complexity too much.

*4.5. Changes to CNNs*

The input for all the studied CNNs requires 3 image bands. Our images consist of only one band. We modified the dataset, transforming it to 3 channels by simply replicating the existing channel. In addition, the input resolution should be higher than the resolution of the images in our dataset. For InceptionV3, the resolution is (299, 299, 3) with a minimum of (75, 75, 3). The initial two values denote the image's width and height, while the final value indicates the number of bands within the image. Most CNN architectures work on images containing three bands that correspond to the three primary colors: red, green, and blue. For Xception, it is (299, 299, 3) with a minimum of (71, 71, 3). For InceptionResNetV2, it is (299, 299, 3) with a minimum of (75, 75, 3). Finally, for ResNet152V2, the resolution is (224, 224, 3) with a minimum of (32, 32, 3). We scaled up all the images to the minimum resolution required for each CNN.

The flatten layer (see Figure 2) was added to transform the multidimensional output received as an input from the previous layers (coming from InceptionV3, Xception, etc.) into a unidimensional dataset. In our case, this layer transforms the feature maps computed by the previous convolution and pooling layers into the data needed by the Dense layer that follows. The dense layer performs the actual classification of the input data by training a set of parameters using a back-propagation mechanism. In this layer, each neuron is connected to every neuron in the previous layer. The connections have weights that are learned during training. To reduce the overfitting that could arise, we added a dropout layer that randomly ignores some neurons during training in the forward- and back-propagation mechanisms. In this manner, we tried to minimize the development of dependencies between neurons during the training phase. The last layer, also a Dense layer, is based on softmax and generates the probability of membership for each of the two classes, asteroids and non-asteroids.

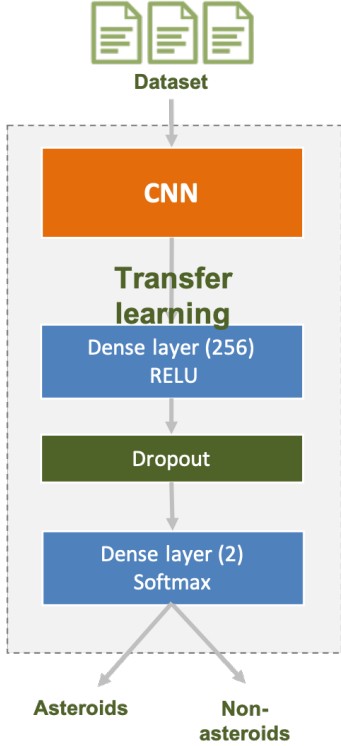

**Figure 2.** Architectural description of the proposed solution.

*4.6. Evaluation Metrics*

Accuracy is widely used with deep CNNs to measure training performance due to its simplicity and relevance in the case of the majority of algorithms. One drawback when using accuracy to compare results from different models is that it is heavily affected by class imbalance that refers to an unequal distribution of different classes or categories within a dataset. In other words, one class has significantly fewer examples compared to the others. For our particular dataset, this could be an issue based on the fact that the dataset is imbalanced. In it, we have more non-asteroid detections than asteroid detections. For this reason, we do not rely only on accuracy as our main metric.

$$Accuracy = \frac{TP + TN}{TP + FP + TN + FN}. \tag{1}$$

TP—true positives, TN—true negatives, FP—false positives, FN—false negatives.

In the context of our experiments, true positives refer to asteroid objects that are correctly classified. True negatives represent non-asteroid objects that are correctly classified. False positives mean non-asteroid objects that are incorrectly classified. Finally, false negatives mean asteroid objects that are incorrectly classified.

One tool that we use to assess the system's performance is the confusion matrix. In a binary classification problem, this is a $2 \times 2$ matrix, where the rows specify the actual input class and the columns specify the predictions of the system. We are more interested in maximizing the number of true positive values rather than maximizing that of true negatives.

Precision is therefore computed as the number of true positives (asteroid objects correctly identified) divided by the sum of true positives and false positives (non-asteroid objects marked incorrectly as asteroid objects). To compute the precision, we take into account just the true positives and the false positives. We exclude the false negatives, which are actually asteroid objects but marked incorrectly by the system. In this way, we obtain high precision if our system has a low number of false positives (meaning that it correctly classifies non-asteroid objects) without taking into consideration the fact that the system misses some asteroid objects. Even though precision is useful, it still misses some important information. It does not incorporate information on how many real positive class examples are predicted to belong to the wrong class. For this reason, we are not relaying on precision when assessing the system performance.

$$Precision = \frac{TP}{TP + FP}. \tag{2}$$

Recall takes into consideration the false negatives and is computed as the number of true positives divided by the sum of true positives and false negatives (asteroid objects marked incorrectly by the system as non-asteroid objects). For this reason, we rely on recall to assess the system performance based on the fact that we want to minimize the number of asteroid objects that are incorrectly classified.

$$Recall = \frac{TP}{TP + TN}. \tag{3}$$

Precision highlights the correct positive predictions out of all the positive predictions, while recall highlights the missed positive predictions. Based on this, recall focuses more on the coverage of the positive class.

The F1 score combines the accuracy and recall metrics into one metric. The F1 score has also been developed to perform effectively on data that is unbalanced. If precision and recall values are both high, a model has a high F1 score. If precision and recall values are both low, a model receives a low F1 score.

$$F1\_score = 2 * \frac{Precision * Recall}{Precision + Recall}. \tag{4}$$

## 5. Experimental Results

### 5.1. Processing Flow

The results presented in this paper are part of the CERES software module for asteroid detection from astronomical images. It is composed of three modules (see Figure 3): one module for creating the image dataset that is used for training a classification model, one module for classifying objects detected from astronomical images for the purpose of identifying asteroids, and the inference module that is integrated into the NEARBY platform.

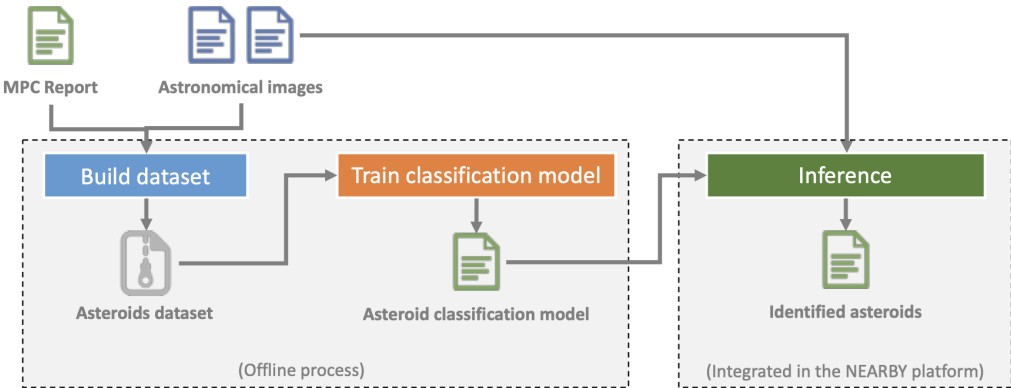

**Figure 3.** Processing flow.

The classification model for this study is trained using Keras [45] and Tensorflow [46] and is implemented in Python. We make use of Astropy [47], a community-developed core Python module for Astronomy, to analyze the FITS astronomical data.

### 5.2. Dataset

The dataset on which we train the asteroid classification model is a binary one, with just two classes: one for valid asteroids (some examples are in Figure 4), and another one for objects that were identified in the input astronomical images but which are not asteroids (some examples are in Figure 5).

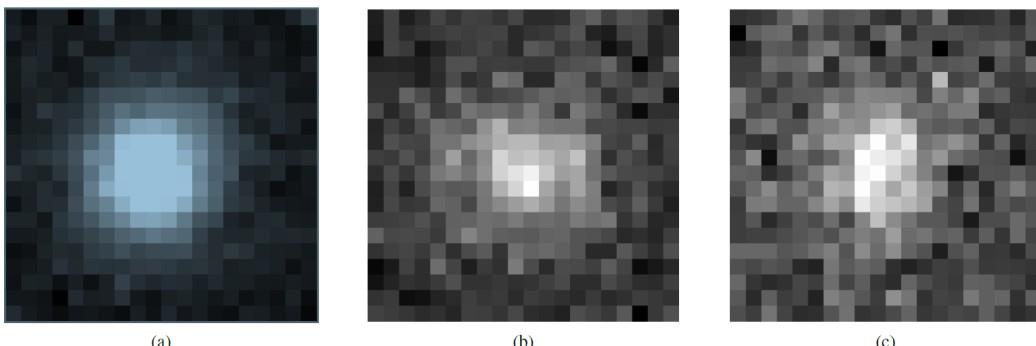

**Figure 4.** Examples of images containing valid asteroids (**a**–**c**).

In the NEARBY project [48], we develop an effective software platform that achieves very fast data reduction of astronomical images. This is quite useful for rapid detection and validation of asteroids. It also offers the option for visual analysis of the processed images and human validation of the moving sources (asteroids), assisted by static and dynamical presentations. The processing and detection are performed over a high-performance computing solution based on a cloud infrastructure. Astronomical images represent the input data for this software platform. Such images are captured using mosaic cameras (containing an array of CCDs (ChargedCoupled Device)) and represent the raw input data for every asteroid detection algorithm. However, before being actually useful, we need to transform and modify them. This process is known as image reduction, and it removes

instrumental signatures, masks cosmic rays and applies photometric and astrometric calibration operations. Next, we need to identify real objects in the astronomical images and try to group them across several images in order to identify an asteroid trajectory.

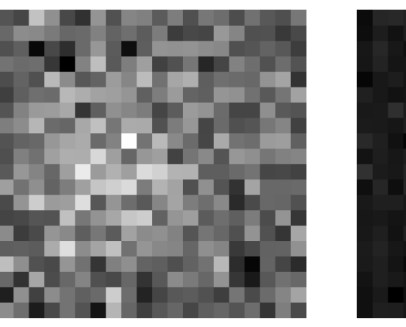 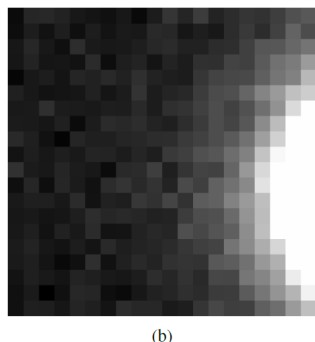 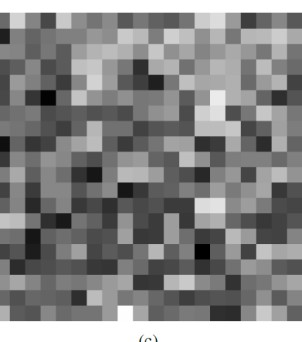

|            |            |            |
|:----------:|:----------:|:----------:|
| (a)        | (b)        | (c)        |

**Figure 5.** Examples of images containing non-asteroids (**a**–**c**).

SExtractor [49] is used to build up a catalogue of objects from astronomical images. The asteroid detection algorithm (CrossObject) that is developed in the NEARBY project relies on the objects extracted from astronomical images by the SExtractor library. As the performed analysis is based solely on the images, without any knowledge of the context, some of the identified objects are erroneous (generally noise) and favor the detection of false positives (incorrectly detected asteroids). Due to this, some of these detections are interpreted as valid trajectories by the asteroid detection algorithm. This type of errors could be relatively easily avoided if we added one more layer between the object extraction phase and the asteroid detection algorithm. The development of this layer, based on machine learning techniques, is being conducted as part of the CERES project [2].

One of the major challenges in successfully applying machine learning techniques within any use-case is the availability of valid training data. In our scenario, the required dataset is built by relying on detections that were already classified and validated by human experts. The input data for the CERES software module comes in the FITS format (Flexible Image Transport System), which is an open standard used to describe a digital file format useful for the storage, transmission and processing of data. Typically, the images contain an implicit Cartesian coordinate system that describes the location of each pixel. However, astronomic images are enhanced with a world coordinate system such as the Celestial coordinate system. In order to build the dataset, the already classified detections are cropped from each astronomical image in which they were reported. The archive of astronomical images to which we have access from our previous work consists of several surveys, and for each survey, we have at least 3 or 4 images with the same telescope orientation (pointing), meaning that each detected asteroid is present in at least 3 or 4 images. Each of these cropped images is part of the new dataset of asteroid images.

The dataset is originally saved in the FITS format, which allows for the specification of pixel values outside of the range of [0, 255]. The pixel data actually indicates light intensity. It is challenging for neural networks to learn because of this wide range of values. We normalize these data to tackle this issue, and the images that result have pixels with values in the range of [0, 1]. We could use data augmentation to increase the dataset. This method is applied in machine learning when the neural network only has a little quantity of data to work with. The goal of augmentation is to provide the network with extra data for training purposes. To obtain additional information, we flip each image vertically or horizontally, rotate the images clockwise or counterclockwise, or apply filters. These methods contribute to the spread of errors if the initial dataset is not sufficiently varied.

Table 1 presents the distribution of asteroids/non-asteroids in the three datasets that we employ: training set, validation set and test set. The distribution of images in the two classes is not uniform, with the non-asteroid class having more images than the asteroid class. We use class weights to aid the model's learning from the unbalanced data.

**Table 1.** Distribution of the images.

|  | **Non-Asteroid Class** | **Asteroid Class** |
|---|---|---|
| Training set | 38,146 | 21,733 |
| Validation set | 433 | 747 |
| Test set | 5983 | 2331 |

Although the images initially had a dimension of $32 \times 32$ pixels, after adding the two additional channels and applying a scale-up, they became too large to be stored directly in the system memory. To tackle this problem, we implement a batch reading component in order to be able to read them from the disk in preparation for the training phase.

*5.3. Experiment Description*

During this study, we performed three experiments as described below.
*Experiment 1:*

(a)    start from the weights calculated using the ImageNet dataset;
(b)    remove the top layer and add some additional layers;
(c)    train the new model.

*Experiment 2:*

(a)    start from the weights calculated using the ImageNet dataset;
(b)    remove the top layer, select a different bottleneck layer and add some additional layers;
(c)    train the new model.

*Experiment 3:*

(a)    start from the weights calculated using the ImageNet dataset;
(b)    remove the top layer, select a different bottleneck layer and add some additional layers;
(c)    unfreeze some of the base model layers and train the new model.

*5.4. InceptionV3*

The first experiment is based on the InceptionV3 model. The results are presented in Tables 2 and 3. We started off by using the ImageNet pre-trained weights. We excluded the fully connected layers from the top of the model. The input should have three channels and at least $75 \times 75$ pixels. The default resolution for the input was $299 \times 299$. Our dataset consisted of images with a lower resolution, just $32 \times 32$ pixels, so in order to accommodate them, we had to scale up all the images. We added some more layers to this model: a flatten layer, a dense layer, a dropout layer and the final layer based on the softmax activation function. The accuracy of the model was 0.87 and the loss was 0.34. This means that, without making any adjustments on the model, we obtained a decent performance. Looking at the recall metric, we could observe that it was similar for both classes. For the asteroid class, we could observe that the model missed quite a lot of valid detections. This suggested that the model is not very good and that we need to make some adjustments.

Next, we added our new layers to the output of the mixed_2 layer of the InceptionV3 model. The accuracy of this model increased to 0.92, while the loss decreased to 0.21. This improvement in performance occurred due to the fact that we used as a bottleneck a layer that had a higher dimensionality. This means that we had much more features than would have been useful to the model to learn from. Also worth mentioning is the fact that in this model, the base layers were frozen and we only trained the new layers that were added.

The last experiment using this model involved the unfreezing of some of the base model (from InceptionV3) and training those layers together with the layers that were added to the model. By unfreezing, we allowed the update of the parameters of a previously frozen model during further training. We unfroze starting from different layers (mixed_0 and mixed_1), but the results were approximately the same. We obtained an accuracy of 0.92 and a loss of 0.22. Even though the recall for non-asteroid objects dropped by some

percentage, the recall for asteroid objects increased. This is the best result for this model (InceptionV3) since we are missing only 112 valid detections from a total of 233.

**Table 2.** InceptionV3 classification results.

|  |  | Precision | Recall | F1-Score |
|---|---|---|---|---|
| Experiment 1 | Non-asteroid class | 0.94 | 0.88 | 0.91 |
|  | Asteroid class | 0.73 | 0.85 | 0.79 |
| Experiment 2 | Non-asteroid class | 0.97 | 0.92 | 0.94 |
|  | Asteroid class | 0.82 | 0.92 | 0.86 |
| Experiment 3 | Non-asteroid class | 0.98 | 0.91 | 0.94 |
|  | Asteroid class | 0.81 | 0.95 | 0.87 |

**Table 3.** InceptionV3 confusion matrix.

|  |  | Predicted Non-Asteroid Class | Predicted Asteroid Class |
|---|---|---|---|
| Experiment 1 | Actual non-asteroid class | 5266 | 717 |
|  | Actual asteroid class | 359 | 1972 |
| Experiment 2 | Actual non-asteroid class | 5507 | 476 |
|  | Actual asteroid class | 192 | 2139 |
| Experiment 3 | Actual non-asteroid class | 5456 | 527 |
|  | Actual asteroid class | 112 | 2219 |

*5.5. Xception*

We started by using the ImageNet pre-trained weights. We excluded the fully connected layers from the top of the model. The input should have three channels and at least $71 \times 71$ pixels, smaller than in the Inception scenario. The default resolution for the input is $299 \times 299$. We had to scale down all the images to a $71 \times 71$ pixel resolution. The results are presented in Tables 4 and 5. The accuracy of this model (off the shelf, no fine-tuning) was 0.86 and the loss was 0.37. These values are very similar to the ones obtained using InceptionV3. In this situation, we obtained a relatively high precision for non-asteroid objects when compared to the precision of the other class. The important aspect are the recall values, which are also very similar to those of InceptionV3. The model wrongly identified a lot of asteroid objects, so we needed to make some improvements in order to increase the recall.

By selecting as bottleneck layer add_1, we achieved an increase in accuracy and a decrease in loss. The new values are 0.90 for accuracy and 0.28 for loss. The precision increases for both classes and we can observe that the recall increases much more on the asteroid objects class than on non-asteroid objects class. With a recall of 94%, this proves to be a good model. We missed only 144 valid detections out of a total of 2331 (6%).

**Table 4.** Xception classification results.

|  |  | | Xception | |
|---|---|---|---|---|
|  |  | Precision | Recall | F1-Score |
| Experiment 1 | Non-asteroid class | 0.93 | 0.86 | 0.90 |
|  | Asteroid class | 0.70 | 0.84 | 0.77 |
| Experiment 2 | Non-asteroid class | 0.97 | 0.90 | 0.93 |
|  | Asteroid class | 0.78 | 0.94 | 0.85 |
| Experiment 3 | Non-asteroid class | 0.98 | 0.90 | 0.93 |
|  | Asteroid class | 0.78 | 0.95 | 0.85 |

**Table 5.** Xception confusion matrix.

| | | Members of Predicted Non-Asteroid Class | Members of Predicted Asteroid Class |
|---|---|---|---|
| Experiment 1 | Actual non-asteroid class | 5147 | 836 |
| | Actual asteroid class | 366 | 1965 |
| Experiment 2 | Actual non-asteroid class | 5362 | 621 |
| | Actual asteroid class | 144 | 2187 |
| Experiment 3 | Actual non-asteroid class | 5356 | 627 |
| | Actual asteroid class | 123 | 2208 |

If we unfreeze some of the bottom layers and keep the bottleneck layer as add_1, we will succeed in decreasing the loss down to 0.27, whereas the accuracy will remain the same, at 0.90. There is also a small improvement in the recall rate, up to 0.95, meaning that we now only miss 123 valid detections out of 2331. For the Xception model, this is the best result that we have achieved.

*5.6. InceptionResNetV2*

Like in the case of previous models, we also started from ImageNet pre-trained weights. The default resolution for the input was $299 \times 299$, with at least $75 \times 75$ and three input channels. The results are presented in Tables 6 and 7. Without any modifications on the model, we achieved a lower loss (just 0.48) compared with the previous models, and an accuracy of 0.84. The precision for the non-asteroid class was just 0.9, versus 0.7 for the asteroid class. We missed a lot of valid detections (588 out of 2331), but we had decent predictions for the non-asteroid class.

The next experiment involved selecting block35_10_mixed as the bottleneck layer. We achieved an increase in accuracy (up to 0.92) and a decrease in loss (down to 0.23). The recall values for asteroid objects increased but were still lower than the recall for the non-asteroid class (0.91 compared to 0.93). This model missed 218 out of a total of 2331 valid detections (9%).

**Table 6.** InceptionResNetV2 classification results.

| | | Precision | Recall | F1-Score |
|---|---|---|---|---|
| Experiment 1 | Non-asteroid class | 0.90 | 0.88 | 0.89 |
| | Asteroid class | 0.70 | 0.75 | 0.72 |
| Experiment 2 | Non-asteroid class | 0.96 | 0.93 | 0.94 |
| | Asteroid class | 0.83 | 0.91 | 0.86 |
| Experiment 3 | Non-asteroid class | 0.96 | 0.93 | 0.94 |
| | Asteroid class | 0.83 | 0.91 | 0.86 |

**Table 7.** InceptionResNetV2 confusion matrix.

| | | Members of Predicted Non-Asteroid Class | Members of Predicted Asteroid Class |
|---|---|---|---|
| Experiment 1 | Actual non-asteroid class | 5236 | 747 |
| | Actual asteroid class | 588 | 1743 |
| Experiment 2 | Actual non-asteroid class | 5540 | 443 |
| | Actual asteroid class | 218 | 2113 |
| Experiment 3 | Actual non-asteroid class | 5541 | 442 |
| | Actual asteroid class | 208 | 2123 |

By unfreezing layers starting from mixed_5b and running the training on all of these layers, we achieved a decrease in loss to 0.2, while the accuracy remained the same, at 0.92. The percentages of recall are the same as for the previous model, but looking at the actual numbers, we can see that we correctly predicted 10 more valid detections.

### 5.7. ResNet152V2

The results are presented in Tables 8 and 9. For ResNet152V2, we obtained the worst result by starting from the pre-trained weights. In this situation, the accuracy was just 0.82, with a relatively high loss of 0.48. The precision for the asteroid objects class was pretty low, just 0.63 with a recall of 0.83. A lot of valid detection would be missed if we were to employ this model.

We managed to achieved a decent improvement by selecting conv3_block8_1_relu as the bottleneck layer. Here, the precision rose to 0.91, while the loss dropped to 0.24. The recall values for both classes were around 0.9. We actually gained twice the number of valid detections.

**Table 8.** ResNet152V2 classification results.

|  |  | Precision | Recall | F1-Score |
|---|---|---|---|---|
| Experiment 1 | Non-asteroid class | 0.92 | 0.81 | 0.86 |
|  | Asteroid class | 0.63 | 0.83 | 0.72 |
| Experiment 2 | Non-asteroid class | 0.96 | 0.91 | 0.93 |
|  | Asteroid class | 0.80 | 0.89 | 0.84 |
| Experiment 3 | Non-asteroid class | 0.95 | 0.94 | 0.94 |
|  | Asteroid class | 0.84 | 0.87 | 0.85 |

**Table 9.** ResNet152V2 confusion matrix.

|  |  | Predicted Non-Asteroid Class | Predicted Asteroid Class |
|---|---|---|---|
| Experiment 1 | Actual non-asteroid class | 4861 | 1122 |
|  | Actual asteroid class | 105 | 1926 |
| Experiment 2 | Actual non-asteroid class | 5460 | 523 |
|  | Actual asteroid class | 245 | 2086 |
| Experiment 3 | Actual non-asteroid class | 5606 | 377 |
|  | Actual asteroid class | 313 | 2018 |

For the previous models, if we unfroze some of the layers, we would gain in performance. Here, we opted for unfreezing starting with the conv2_block3_2_conv layer. The accuracy in this situation remained the same, but we achieved a drop in loss (down to 0.21 from 0.24). The downside was that the number of false negatives increased, meaning that we lost some of the valid detections.

## 6. Discussion

In the experiments detailed in this paper, we employed transfer learning and fine-tuning techniques on established deep convolutional networks that were initially pretrained using the ImageNet dataset. Figure 6 presents, by comparison, the confusion matrices for each of the experiments performed on each CNN based architectures. The best results, in terms of recall for the asteroid class, were obtained using InceptionV3 by fine-tuning of some of the parameters. We obtained a recall of 0.95 in this scenario. InceptionResNetV2 has the lower recall due to simply performing transfer learning without training some of the base model layers. The performance of the Xception model is similar to that of the InceptionV3 model.

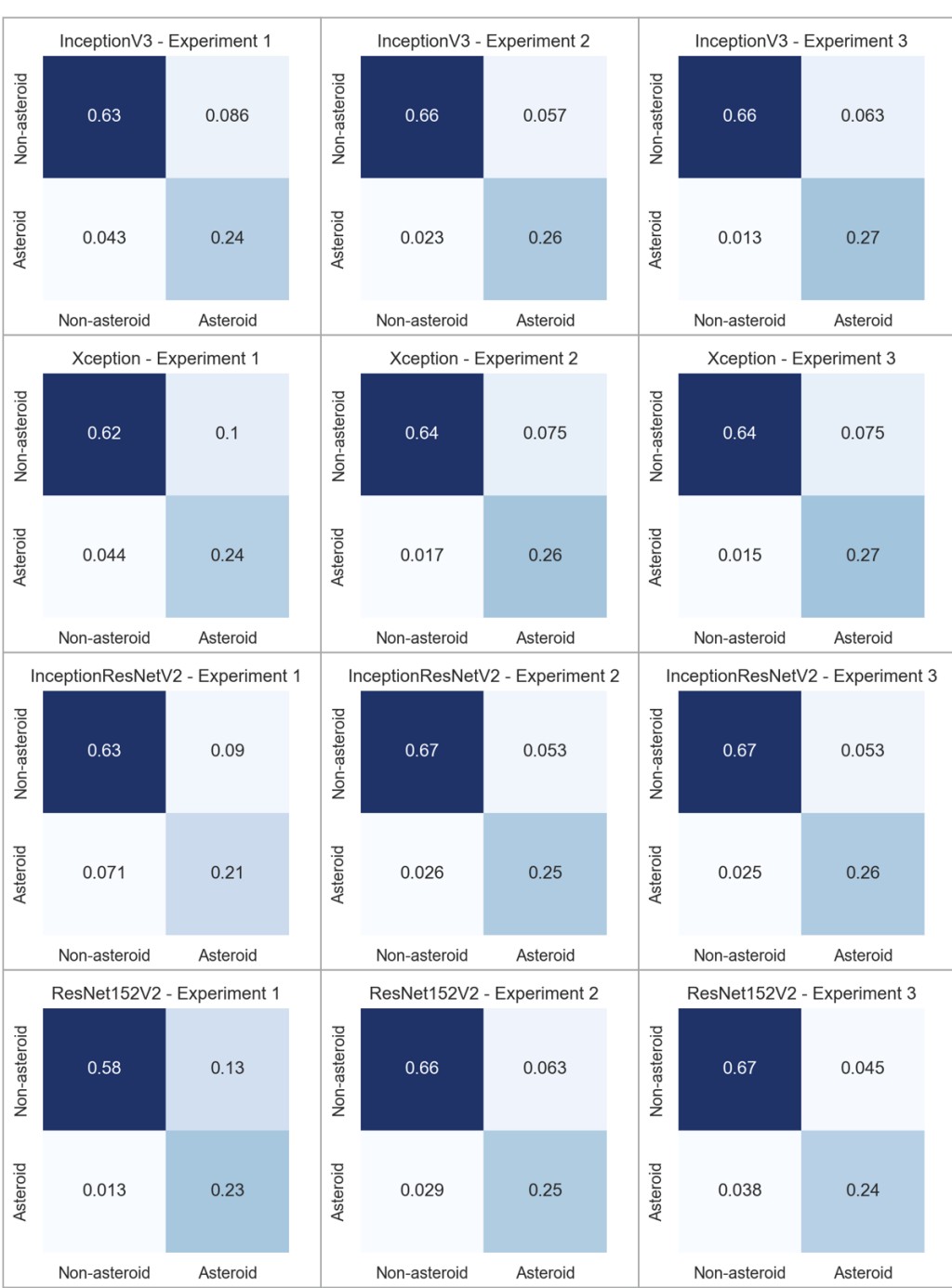

**Figure 6.** Confusion matrices.

Our dataset is very different from the ImageNet dataset that was used for pre-training all of the studied models. ImageNet does not contain these classes. Another difference comes from the fact that the ImageNet dataset has red, green and blue bands instead of just one band, as in our dataset. For this reason, we had to increase the dimensionality of the dataset by duplicating the existing channel. Due to the small dimensions of an asteroid in an astronomical image (number of pixels), the initial resolution of the images in the dataset was quite low ($32 \times 32$) compared to the values needed for the CNNs models. For this reason, we had to scale up all the images. This scaling operation could also influence the obtained results.

However, from the results that we obtained (Figure 7), one can see the potential of using deep convolutional neural networks in the process of asteroid classification. Coupled

with other techniques and technologies, like distributed computing, they could help in greatly reducing the amount of data that human experts have to sift through in order to find valid asteroids.

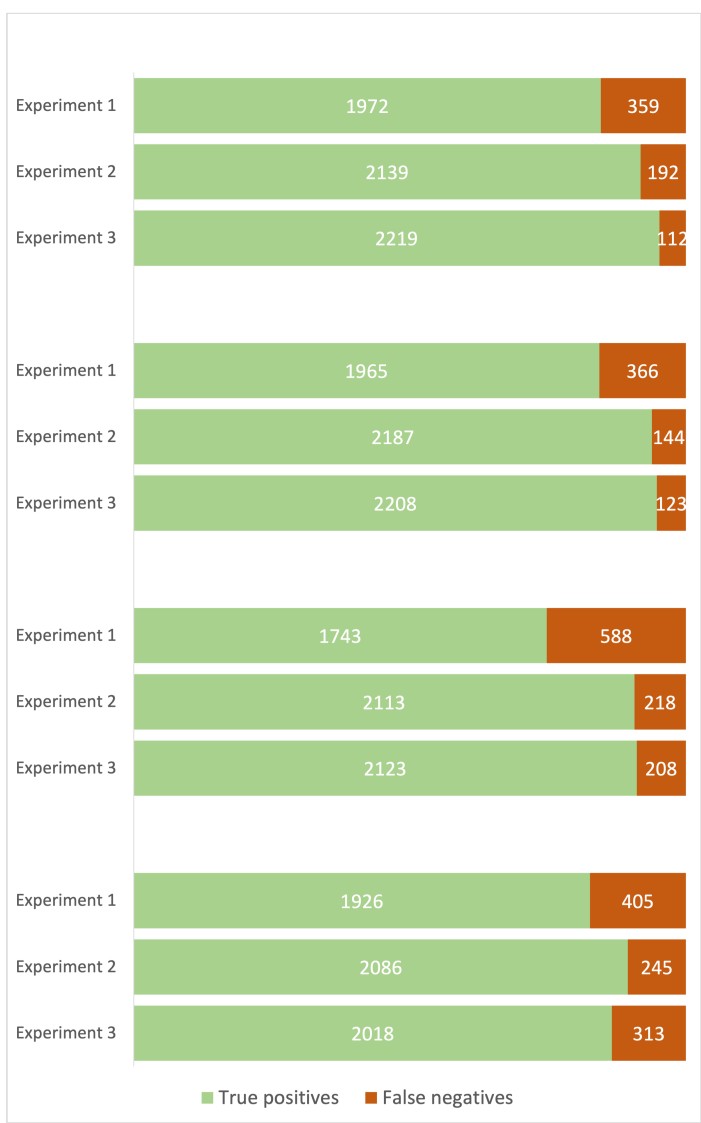

**Figure 7.** Correct vs. incorrect classification for the asteroid class.

## 7. Conclusions

The main goal of the present paper was to assess the effectiveness of employing deep CNNs for classifying astronomical objects, in particular, asteroids. In our case, the problem was that of a binary classification, relegating the identified objects to one of two possible classes: asteroids and non-asteroids. We compared some of the most well-known deep CNNs, including InceptionV3, Xception, InceptionResNetV2 and ResNet152V2. The results of this study highlight the fact that the high-level features learned by deep neural networks are effective for the classification of asteroids. The InceptionV3 model has the best results in the asteroid class, meaning that by using it, we lose the least number of valid asteroids. Overall, we are of the opinion that the results obtained and documented herein demonstrate the potential of employing deep CNNs in solving the problem of classifying astronomical objects.

**Author Contributions:** V.B.: Conceptualization, Methodology, Software, Validation, Writing. C.N.: Software, Validation, Writing. A.S.: Software, Validation, Writing. T.S.: Software, Validation, Writing. D.G.: Conceptualization, Methodology, Validation. All authors have read and agreed to the published version of the manuscript.

**Funding:** This work was supported by a grant of the Romanian Ministry of Education and Research, CCCDI—UEFISCDI, project number PN-III-P2-2.1-PED-2019-0796, within PNCDI III (the development of the dataset and CNN models). This research was supported by the CLOUDUT Project, cofunded by the European Fund of Regional Development through the Competitiveness Operational Programme 2014–2020, contract no. 235/2020 (the Cloud infrastructure that was used in order to train the CNN models). The data used to train and test the model developed as part of this research effort were made available by observations performed with the Isaac Newton Telescope (INT), operated on the island of La Palma by the Isaac Newton Group (ING) in the Spanish Observatorio del Roque de los Muchachos (ORM) of the Instituto de Astrofisica de Canarias (IAC). The work was also supported by the project "Entrepreneurial competencies and excellence research in doctoral and postdoctoral programs—ANTREDOC", a project co-funded by the European Social Fund, financing agreement no. 56437/24.07.2019. This research was partially supported by the project 38 PFE in the frame of the programme PDI-PFE-CDI 2021.

**Data Availability Statement:** Not applicable.

**Conflicts of Interest:** The authors declare no conflict of interest.

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
