# Peer review of "Assessment of Asteroid Classification Using Deep Convolutional Neural Networks"

_aerospace, doi:10.3390/aerospace10090752_

Round 1

Reviewer 1 Report

This is an interesting paper describing application of convolutional neural networks to detecting asteroids from astronomical imagery. The paper appears sound with no problems except the very minor issues listed below.

Line 86. No need to define CNNs again since it was already defined on line 54. Consider moving that definition to the first use, which is in the abstract, and/or line 51 for the main text.

Line 247. Spelling of “fine-tuning”

Eq. 1. I can guess what TP, TN, FP, and FN mean, but I think it might be helpful to define them for readers who may not be able to guess so quickly. For example, in line 334 you might put “(TP)” after “true positives” in the text, etc.

Reviewer 2 Report

General aspects

The manuscript is about the evaluation of convolutional neural network based automatized asteroid identification methods. The topic is relevant, both regarding asteroid identification and the neural network based methodological approach. The structure of the manuscript is moderately good, the language is very good, there are new results obtained. The context is moderately well described, but some further links and references are needed. However there are several basic problems that should be corrected. The main issues are the followings:

·       The basics are not well described, some very simple statements are missing from the Introduction. Please indicate that for asteroid identification the main point is to separate the moving objects (visible in different images) from non-moving stars. This should be stated.

·       Indicate what are the main criteria for classifying an object as asteroid.

·       The used “classification” term is good, however at the beginning it should be noted that here the “identification” of asteroids and their “separation” from the stars is the aim.

·       There are several further terms, which should be briefly explained, see among the specific suggestions below.

·       Some information is also needed to lay down the stage (science case as reason to use the convolutional neural network), suggest to indicate in the introduction that asteroids are being discovered by sky surveys in large numbers recently (https://ui.adsabs.harvard.edu/abs/2023PSJ.....4..128G/abstract), recently used automatized classification about asteroid spectral classes (https://ui.adsabs.harvard.edu/#abs/2021A%26A...649A..46P/abstract), correlating to meteorite laboratory spectra (https://ui.adsabs.harvard.edu/abs/2020P%26SS..18404855S/abstract), with automatized methods (especially important for Earth asteroids (https://ui.adsabs.harvard.edu/abs/2021A%26A...649A..46P/abstract), which have impact risk (https://ui.adsabs.harvard.edu/#abs/2023Sci...379.1179V/abstract).

·       Size does matter much for asteroid brightness, would be useful to indicate the role of observed brightness in the asteroid identification, especially approaching the limiting magnitude.

Specific aspects

3-4 lines

objects that pass through the Earths vicinity.

OK, but asteroid identification is also relevant for more distant objects, actually main belt asteroids are much more numerous than NEAs, so the identification / classification activity is also releavnt (and even more frequent) for main belt asteroids

19

their proximity to the Earth

also mention with the possibility of impact with the Earth

40

It is very important

suggest to delete very

47

dataset of asteroid images

does it mean series of subsequent images of the same asteroids?

49

asteroid classification

this is not the proper term and formulation. Asteroid classification is used in the literature for classification of known asteroids between different classes (e.g. S type, C type etc.), here this work is about selection of asteroids from stars or other objects and their identification. Classification is not the proper term.

50

accomplish the inference

not clear what do the authors mean

78

supervised and unsupervised learning problems

if there is a work to cite, please indicate

95

machine learning techniques

also cite: https://ui.adsabs.harvard.edu/#abs/2023JGCD...46.1280T/abstract

110-111

suggest to mention that automatized identification of transient objects like supernovae, novae, variable stars, as well as artificial satellites that also need to be separate from all other objects

117

identifying such small planets mainly

not clear, do you mean transit related brightness decrease for exoplanet identification?

around 170-180

indicate for asteroid identification the movement between subsequent images compared to the “standing” stars does matter

188

softmax function

cite relevant reference

192

geometrical representation of an asteroid and the patterns that are specific to asteroids.

not clear what do you mean, motion of asteroids?

194

low level features

explain briefly

edges or corners

of what?

200

Rectified Linear Unit (ReLU).

is there citation for this?

208

At the same time, the relevant information (features) are retained.

not clear, please explain differently

207

like maximum or average

suggest to complete to like maximum or average calculation

213

overfitting.

explain briefly

214-216

please reformulate this part, difficult to understand

234

relatively small

the number or memory size? if possible please quantify or compare to something

245

Full training

247

fine-tunning approach

explain brielfy both

around 7 of 20 page

explain the role of connection(s) between different layers briefly

292

residual connections

please explain briefly

307

3 image bands.

explain what are these bands

310

resolution is (299, 299,3)

explain for non experts what does it mean

324

Dense layer

explain for non expert what is it for

330

class imbalance

explain for non-experts

333 (1) fromulae

explain the acronyms, where TP is for…”

361

F1 score

explain what is it

384

moving sources

more info is needed on the motion identifiation, it is very important

Figure 5

suggest to put here regular star image also

406

availability of valid training data

suggest to give more info on it

around 410

more info on the spatial accuracy would be useful on idetified coorinates or other mean of apatial loction

Table 1

why are there only a bit more non-asteroid than asteroids? Not all stars were counted?

433

images initially had a dimension of 32x32 pixels

were this the size of the whole images? whole images should have been several orders of magnitude larger in pixel size than indicated

468

The accuracy of this model increased to 0.92

some more explanation would be useful on what this values mean

475

unfroze starting

give some further info on this for non-experts

Table 5 and 7

the title of the last two columns might becompleted with members of predicted…”

495

only 144 valid detections out of a total of 2331

give in % also

also in 512: 218 out of a total of 2331, and 617: 10 more valid detections

529

conv2_block3_2_conv

it would be quite useful if there was a list of mentioned specific layers, indicated their tole and connection with others to see the whole system and interactions. This might be supporting online material.

547

Due to the small dimensions of an asteroid

please clarify this, the specific pixel size? and what about various brightness star sizes on images?

553

techniques an technologies

modify to and

558

classifying astronomical objects

do you mean to classify to asteroids, stars, various dim objects (clouds, galaxies) and artificial satellites?

564

we lose the least

modify to loose

around the end of the conclusion some suggestions toward future use or further steps would be important

The language is OK.

Author Response

Thank you for the very useful comments. Please see the attachment. 

Round 2

Reviewer 2 Report

Thank you for the revission, the manuscript is almost ready for publication. Last requests:

- please refomulate the line 268, a bit difficult to understand

- 330 line: please explain in brackets "number of bands"

- a final language checking would be useful

The English is moderately good.